# Assessment of Functional and Nutritional Status and Skeletal Muscle Mass for the Prognosis of Critically Ill Solid Cancer Patients

**DOI:** 10.3390/cancers14235870

**Published:** 2022-11-29

**Authors:** Clara Vigneron, Othmane Laousy, Guillaume Chassagnon, Maria Vakalopoulou, Julien Charpentier, Jérôme Alexandre, Matthieu Jamme, Frédéric Pène

**Affiliations:** 1Service de Médecine Intensive-Réanimation, Hôpital Cochin, Assistance Publique-Hôpitaux de Paris, Centre-Université Paris Cité, 75014 Paris, France; 2Université Paris Cité, 75014 Paris, France; 3Service de Radiologie, Hôpital Cochin, Assistance Publique-Hôpitaux de Paris, Centre-Université Paris Cité, 75014 Paris, France; 4Ecole CentraleSupelec, Mathématiques et Informatique pour la Complexité et les Systèmes, Université Paris-Saclay, 91190 Gif-sur-Yvette, France; 5Service D’oncologie Médicale, Hôpital Cochin, Assistance Publique-Hôpitaux de Paris, Centre-Université Paris Cité, 75014 Paris, France; 6Service de Médecine Intensive-Réanimation, Hôpital Privé de l’Ouest Parisien, Ramsay Générale de Santé, 78190 Trappes, France; 7Team 5 (EpReC, Renal and Cardiovascular Epidemiology), Centre de Recherche en Epidémiologie et Santé des Populations, Université de Versailles Saint-Quentin, 78000 Villejuif, France; 8INSERM U1016, CNRS UMR8104, Institut Cochin, 75014 Paris, France

**Keywords:** cancer, ICU, outcome, skeletal muscle mass, albumin

## Abstract

**Simple Summary:**

Critically ill cancer patients requiring intensive care unit (ICU) admission remain affected by high mortality rates. We wanted to evaluate how their ability to perform daily activities, their nutritional status, and muscle surface can impact their prognosis. Muscle surface was determined through automatic analysis of computed tomography scans using a deep learning-based technique. We found that most cancer patients displayed low muscular surface upon ICU admission, and that muscle surface, the ability to perform daily activities, and the six month course of nutritional parameters were able to predict short- and long-term prognosis.

**Abstract:**

Simple and accessible prognostic factors are paramount for solid cancer patients experiencing life-threatening complications. The aim of this study is to appraise the impact of functional and nutritional status and skeletal muscle mass in this population. We conducted a retrospective (2007–2020) single-center study by enrolling adult patients with solid cancers requiring unplanned ICU admission. Performance status, body weight, and albumin level were collected at ICU admission and over six months. Skeletal muscle mass was assessed at ICU admission by measuring muscle areas normalized by height (SMI). Four-hundred and sixty-two patients were analyzed, mainly with gastro-intestinal (34.8%) and lung (29.9%) neoplasms. Moreover, 92.8% of men and 67.3% of women were deemed cachectic. In the multivariate analysis, performance status at ICU admission (CSH 1.74 [1.27–2.39], *p* < 0.001) and the six month increase in albumin level (CSH 0.38 [0.16–0.87], *p* = 0.02) were independent predictors of ICU mortality. In the subgroup of mechanically ventilated patients, the psoas SMI was independently associated with ICU mortality (CSH 0.82 [0.67–0.98], *p* = 0.04). Among the 368 ICU-survivors, the performance status at ICU admission (CSH 1.34 [1.14–1.59], *p* < 0.001) and the six-month weight loss (CSH 1.33 [1.17–2.99], *p* = 0.01) were associated with a one-year mortality rate. Most cancer patients displayed cachexia at ICU admission. Time courses of nutritional parameters may aid the prediction of short- and long-term outcomes.

## 1. Introduction

Solid cancer patients account for one in ten intensive care unit (ICU) admissions; this proportion has increased over the last decade [1,2]. Avoiding futile care while also avoiding the tendency to excessively deny admission is crucial in this population [3]. Moreover, given the particular prognostic factors related to malignancy and the patient’s preferences, simple and accessible prognostic factors are paramount in the decision-making process when discussing admission policies [4]. Organ failure support during an ICU stay drives the short-term prognosis, but this is hardly predictable before ICU admission. Oncologic characteristics such as type of cancer, stage, and status of response to antitumoral treatment have little impact on short-term survival, but they are strongly associated with one-year survival [2,3]. In addition, the functional status, as assessed by the simple and reproductive performance status scale, is now fully recognized as a strong prognostic factor in critically ill cancer patients. Indeed, severely impaired functional status (performance status 3–4 meaning bedridden for more than 50% of waking hours) has been consistently associated with death in several studies [5,6,7,8,9]. 

Functional status relies on disease stage but also on nutritional status and muscle preservation. A low body mass index (BMI) has been associated with worse short and long-term outcomes in intensive care patients, including 20% of patients with cancer, and mostly those who have undergone major elective surgery [10]. Blood markers such as albumin and prealbumin can predict outcomes in cancer patients undergoing surgery [11,12]. The dynamic course of those parameters over the months prior to ICU admission may add to the assessment of patients’ physiological reserves and their further capacity for recovery. Reduced muscle mass, as screened by computed tomography (CT) measurements, has emerged as a potent prognostic marker in non-critically ill solid cancer patients and after elective pneumonectomy surgery [13,14,15,16]. An accurate quantification of skeletal muscle mass can be achieved through a CT-scan deep-learning method assessment of the muscle area. 

Our study aims to evaluate the respective impacts of functional and nutritional parameters with a particular focus on muscle mass in critically ill solid cancer patients who required an unplanned ICU admission. 

## 2. Materials and Methods

### 2.1. Study Design and Patients Cohort

We performed a retrospective single-center study that enrolled consecutive solid cancer adult patients requiring an unplanned admission to a 24 bed medical ICU over a 14 year period (2007–2020). Post-operative management is usually provided by an independent surgical ICU. Exclusion criteria included patients who were admitted to secure a procedure and planned admission after elective surgery. Some data related to this cohort were previously reported elsewhere [2,17].

### 2.2. Data Collection

The following data related to the underlying malignancy were collected: date of diagnosis, primary tumor site, staging (localized, advanced, metastatic), oncological status according to RECIST (newly diagnosed within one month before ICU admission, partial remission including stable disease, complete remission, progression), and antitumoral treatment within the last three months (cytostatic chemotherapy, targeted therapy, immunotherapy, radiotherapy, surgery).

Concerning functional and nutritional status, we collected the following clinical and biological variables: performance status prior to the current acute complication and during the preceding six months, BMI [weight (kg)/height (m)^2^] at ICU admission, weight variations within the preceding six months, and albumin and prealbumin levels at ICU admission and in the preceding six months. Performance status was scored at 3 if the patient was only capable of limited self-care and confined to a bed or chair for more than 50% of waking hours and at 4 if completely disabled and totally confined to bed or chair.

Causes for ICU admission were classified into specific when directly linked to manifestations of malignancies, non-specific including toinfection and bleeding and related to therapeutic adverse events such as drug-related side-effects or procedural complications. The severity at ICU admission was assessed by the Sequential Organ Failure Assessment (SOFA) score [18]. Organ failure supports initiated during the ICU stay were recorded, including invasive and non-invasive mechanical ventilation, vasopressor/inotrope support, and renal replacement therapy. Decisions to forgo life-sustaining therapies were collected. The main outcomes were in-ICU and one-year mortality and duration of invasive mechanical ventilation.

### 2.3. Muscle Mass Assessment

Abdominal-pelvic CT scans performed at the time of ICU admission (within one month before, or within, the first week after ICU admission) were analyzed automatically using a deep learning-based technique [19]. At first, a deep reinforcement learning algorithm was used to localize the middle of the third lumbar vertebrae (Figure 1). Then, the identified slice was used as an input for a U-Net with 4 depth layers to segment both the psoas and parietal muscles. This end-to-end pipeline has been trained on more than 900 patients, and it achieved state-of-the-art detection results [19].

Based on the obtained segmentations, we measured the psoas muscle area (cross-sectional area of the right and left psoas, cm^2^), the parietal muscle area (cross-sectional area of skeletal muscles excluding psoas, cm^2^), and the total muscle area (sum of psoas and parietal muscle areas) (Figure 1). Measurements were performed using an attenuation threshold of −29 to 150 Hounsfield units to obtain values of pure muscle area excluding fatty degeneration. The skeletal muscle index (SMI) corresponds to the muscle area normalized by height (cm^2^/m^2^). Based on international expert consensus, the lumbar SMI lower than 55 cm^2^/m^2^ for men and 39 cm^2^/m^2^ for women defined cachexia [20]. 

### 2.4. Statistical Analysis

Continuous variables were expressed as medians (interquartile range) and categorical variables were expressed as counts (percentages). The independent predictors of ICU deaths were addressed in a multivariate Cox cause-specific model by performing stepwise regression with bidirectional approach for the covariate selection procedure. The model included variables that reached a *p* value below 0.20 during univariate analysis. The effects of performance status, muscle surface, weight variation, and albumin were evaluated separately because of the potential interactions. Proportional hazard assumptions were graphically checked, and potential interactions were tested in the final model. We performed a similar analysis for the determinants of one-year mortality in ICU survivors. Decisions to forgo life-sustaining therapies were not entered into the model for ICU mortality, but they were entered into the model of one-year mortality in ICU survivors. All tests were two sided, and *p* values < 0.05 were considered statistically significant. All analyses were carried out using R 3.5.1 and R Studio (R foundation for Statistical Computing, Vienna, Austria). 

### 2.5. Ethics Approval

The study was approved by the ethics committee of the French Intensive Care Society (Société de Réanimation de Langue Française, #CE SRLF 17–03) which waived the need for signed consent, in accordance with French regulations.

## 3. Results

The study comprised 462 solid cancer patients who required an unplanned ICU admission (see Appendix A). The patients’ characteristics are displayed in Table 1. Their median age was 64 years (56–72) and the majority of them were men (64.5%). The gastro-intestinal tract (34.8%) and the lung (29.9%) were the most frequent primary tumor sites. Most patients exhibited advanced malignancies, with 75.1% at a metastatic stage and 50.2% in progression under treatment. The majority of patients were receiving active treatment within the past three months, 75.1% with cytostatic chemotherapy and 28.4% with targeted- or immunotherapy either alone or associated with cytostatic chemotherapy.ICU admissions were warranted mainly because of an infection (38.1%), specific complications linked to the underlying malignancy (22.5%), or therapeutic adverse events (16%). One-hundred and fifty-one (32.7%) patients required invasive ventilatory support during the ICU stay, whereas 29.9% required circulatory support with vasopressors or inotropes, and 11.0% underwent renal replacement therapy.

Functional and nutritional parameters are shown in Table 2. At ICU admission, 15% of patients had severely impaired functional status (performance status 3 or 4) prior to the acute complication (Table 2). Anterior values of the clinical and biological parameters were available for the majority of patients, thereby allowing an accurate assessment of the patients’ fitness levels prior to the acute complication. Thus, in 53.3% and 78.3% of patients, a ten percent reduction in weight and a five percent decrease in albumin were observed in the six months prior to ICU admission, respectively. The skeletal muscle index of the psoas area, parietal muscle area, and total muscle area were 4.4 (3.8–5.9), 35.0 (30.0–41.2), and 39.3 (33.9–46.8) cm^2^/m^2^, respectively. With respect to the defining thresholds of cachexia, the skeletal muscle index of the total muscle area was below 55 cm^2^/m^2^ in 92.8% of men and below 39 cm^2^/m^2^ in 67.3% of women.

The in-ICU, in-hospital, and one-year mortality rates were 20.3%, 23.6%, and 73.0%, respectively. Independent predictors of ICU mortality were assessed using the multivariate Cox cause-specific model. The predictors were as follows (Table 3): surgery in the past three months (cause specific hazard (CSH) 0.15 [0.04–0.58], *p* = 0.0006), specific complications (CSH 2.09 [1.02–4.28], *p* = 0.04), admission SOFA score (CSH 1.21 [1.11–1.31], *p* < 0.001), performance status at ICU admission (CSH 1.74 [1.27–2.39], *p* < 0.001), and a six month increase in albumin level (CSH 0.38 [0.16–0.87], *p* = 0.02). In a subgroup analysis of patients undergoing invasive mechanical ventilation, the SMI psoas area was independently associated with ICU mortality (CSH 0.82 [0.67–0.98], *p* = 0.04) (Table 3).

Among the 368 ICU-survivors, independent factors associated with one-year mortality were lung cancer (CSH 1.84 [1.29–2.64], *p* < 0.001), cancer in progression (CSH 2.24 [1.57–3.18], *p* < 0.001), decisions to forgo life-sustaining therapies (CSH 2.40 [1.65–3.50], *p* < 0.001), performance status at ICU admission (CSH 1.34 [1.14–1.59], *p* < 0.001) and six month weight loss (CSH 1.33 [1.17–2.99], *p* = 0.01) (Table 4).

## 4. Discussion

Due to major therapeutic advances, the overall prognosis of cancer has improved over the last two decades, thus leading to broader ICU admission policies; however, critically ill cancer patients remain affected by high mortality rates. Studies aim to address the prognostic determinants in this setting to offer adapted life support in patients with reasonable chances of recovery, and conversely, to avoid futile ICU admissions. In particular, factors available before ICU admission are required to better anticipate the indications and the levels of care provided for life-threatening complications. The oncologic characteristics are known to have little impact on short-term survival, though they are important determinants of long-term outcomes [2,3]. The extent and reversibility of organ failures cannot be predicted before ICU admission since these factors are highly dependent on the definite diagnosis of the acute complication. Patients’ preferences regarding invasive management in the ICU are highly desirable, but most often, they are not expressed, and their preferences are unlikely to cover all situations encountered. An appraisal of the patients’ functional and nutritional status appears to be essential for the decision-making process. Performance status prior to the acute complication is a potent predictor of ICU and hospital mortality [21]. Severely impaired patients are more likely to have advanced underlying disease and they are also more likely to exhibit more severe complications that lead to ICU admission [22]. In our study, this score was independently associated with ICU mortality, but also long-term mortality, because it could have precluded the continuation of antitumoral treatment in ICU survivors. 

The links between impaired nutritional markers and short or long-term outcomes were limited. Weight loss before ICU admission was associated with one-year mortality in ICU survivors, although the association between BMI and survival is inconsistent across cancer types and stages [23]. Most patients from the present cohort exhibited low albumin and prealbumin levels at admission, and we found an association between a six month increase in albumin level and ICU survival. Lower BMI and albumin and prealbumin levels have been associated with a worse short-term prognosis, mainly in post-operative patients or in non-cancer patients, which is a situation wherein the impact of multiple organ failures might be less pronounced [10,11,12,24]; however, the serum albumin level is neither an accurate nor a specific marker of nutritional status, since it is also influenced by inflammation, intravascular fluid content, drugs, and many other conditions impacting its synthesis or loss [25,26]. In addition to nutritional status, the serum albumin level may thus rather be considered as a surrogate for general health condition. 

Sarcopenia is primarily an aging-related loss of muscle mass and function, but it is not well-defined in the literature since it is likely to be influenced by gender, age, and country of origin [27,28,29]. In 2011, a panel of experts used a Delphi process to propose different definitions of cancer cachexia, including a BMI lower than 20 kg/m^2^, a five percent weight loss over the preceding six months, as well as a low lumbar skeletal muscle index (men < 55 cm^2^/m^2^; women < 39 cm^2^/m^2^) [20]. Using these cut-off values in the present cohort, we observed that 81.3% of cancer patients that were admitted to the ICU were considered cachectic. This proportion is quite similar to that reported in the literature (69.1% of metastatic cancer patients exhibited a low muscle mass) and it is significantly higher than that of non-cancer patients hospitalized in the ICU [30,31].

A CT muscle measurement requires close collaboration with radiologists. We evaluated the muscle parameters via the consensual method at the third lumbar vertebra level, and we performed a new fast and accessible deep-learning method, thus allowing an automatic determination of skeletal muscle mass. Furthermore, recently developed assessments, such as multi-vertebral level skeletal muscle analyses using percentile-based averaging and skeletal muscle gauges, are promising [28,32]. Those new methods allow for the determination of new relevant prognosis factors [33]. CT scans are often performed in this population, and this measurement method is more easily accessible than others [34]. 

Whether reduced muscle mass is associated with a worse prognosis in the ICU is discussed, and it is dependent on the selected population [35]. In a study involving 401 patients from the medical ICU, pectoralis muscle area was independently associated with hospital-survival [36]. According to the current definition, cachexia was particularly frequent in our cohort. Of note, only one third of our patients required mechanical ventilation, a subgroup where the impact of reduced muscle mass might be critical for the weaning process [37,38]. Indeed, the SMI psoas surface was independently associated with in-ICU mortality. This suggests that reduced muscle mass is not only a surrogate marker of weakness that is related to underlying comorbid conditions, but it can also interfere with the course of organ failures [39]. Elfassy et al. recently studied the body composition in a cohort of mechanically ventilated patients with hematologic malignancies [40]. Subcutaneous fat and the fat index were significantly associated with longer mechanical ventilation weaning, but not muscle mass. Muscle mass was evaluated using a less validated measurement from the carinal skeletal muscle cross-sectional area. Concerning long-term prognosis, two studies analyzing other muscular parameters found an association with six month mortality, involving a small proportion of cancer patients [41,42]. Another recent study showed that reduced muscle mass was associated with a worse five year outcome in lung cancer patients undergoing pneumonectomy, but only 15.7% required mechanical ventilation after surgery [43]. We could not find any definite association between muscle mass and long-term prognosis, but the dismal one-year survival rate was already largely explained by the advanced underlying malignancy. 

CT-scan screening for reduced muscle mass can now be conveniently performed using artificial intelligence. It may result in targeted interventions involving nutritional care and physical activity; therefore, the tolerance to cancer treatment could be improved and physiological reserve in case of acute complication could be secured. However, targeted interventions in the ICU are more limited since adapted nutrition and early rehabilitation are already routinely implemented in all critically ill patients, but these interventions can be extended after ICU discharge [44]. Furthermore, following-up with morphometric measurements after a critical illness could indicate the capacity for recovery. Cox et al. showed that two thirds of post-septic patients had persistent acute muscle wasting at three months, particularly in patients with reduced muscle mass at baseline [31]. 

Our study has several limitations. Due to its monocentric design, our results may not be fully transcribable to other centers. A number of data were missing, but two thirds of patients actually had baseline morphometric measurements or prealbumin dosage. Most patients exhibited advanced malignancies that strongly impacted on their one-year prognosis. Beyond crude survival, the quality of life, the optimal continuation of oncological treatment and the cancer response to treatment after ICU discharge are also meaningful parameters possibly linked to nutritional and functional status. We did not address these alternative outcomes. In oncology, reduced muscle mass is associated with a worse quality of life, but also with symptoms of depression [45]. In terms of oncological treatment, it is associated with increased toxicity when receiving cytostatic chemotherapy and a poorer response to immune checkpoint inhibitors [13,46].

## 5. Conclusions

We herein provided a comprehensive assessment of functional, nutritional, and muscular status in critically ill solid cancer patients requiring ICU admission. Most cancer patients displayed cachexia at the time of ICU admission. Time courses of nutritional parameters within the preceding six months may add to the prediction of cancer patient’s short-term and long-term outcome.

## Figures and Tables

**Figure 1 cancers-14-05870-f001:**
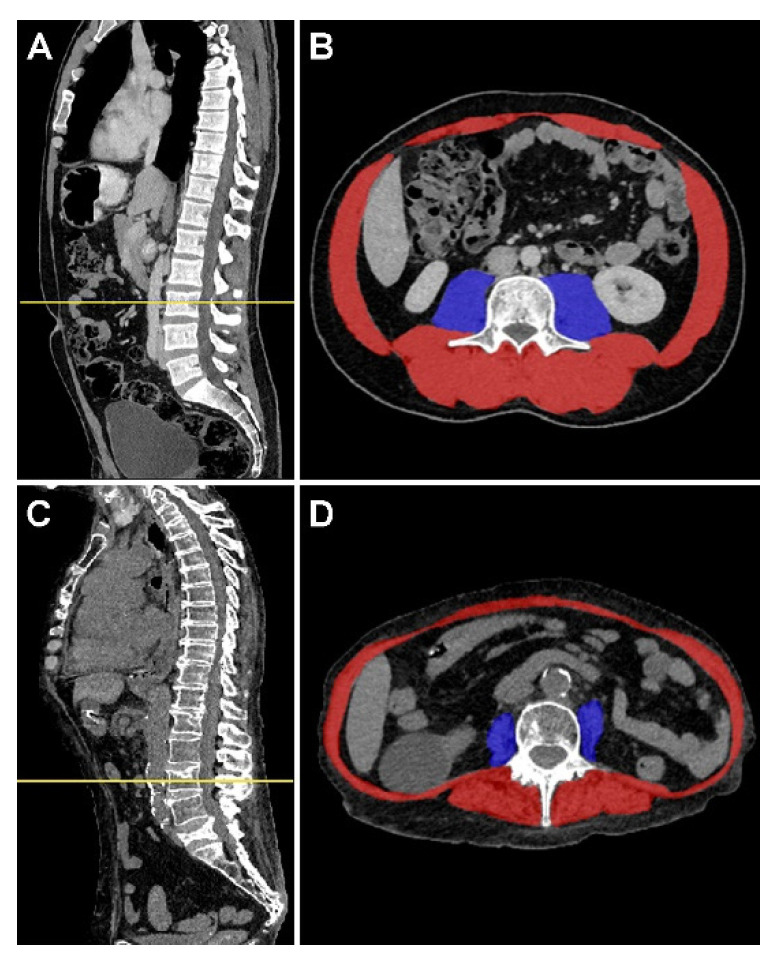
Automatic assessment of the muscle area on the abdominal–pelvic computed tomography scan. Sagittal views show the axial line section in the middle of the third lumbar vertebra using a deep reinforcement learning algorithm (panels (**A**,**C**)). Panels (**B**,**D**) indicate the corresponding axial slice of a fit (**C**) and a sarcopenic (**D**) patient with area measurements of psoas muscle (blue) and parietal muscle (red).

**Table 1 cancers-14-05870-t001:** Patients’ characteristics and outcomes.

Characteristics	n = 462 Patients
Age (years)	64 (56–72)
Male gender	298 (64.5)
Time from cancer diagnosis to ICU admission (days)	273 (113–731)
Type of cancer	
Gastrointestinal	161 (34.8)
Lung	138 (29.9)
Others ^a^	163 (35.3)
Stage	
Localized	25 (5.4)
Advanced	90 (19.5)
Metastatic	347 (75.1)
Cancer status (available for 458 patients)	
Newly diagnosed	89 (19.3)
Partial remission	128 (27.7)
Complete remission	9 (1.9)
In progression	232 (50.2)
SOFA score at ICU admission (points)	5 (4–7)
Organ failure supports	
Invasive mechanical ventilation	151 (32.7)
Duration of mechanical ventilation (days)	3 (2–7)
Non-invasive mechanical ventilation	56 (12.1)
Vasopressors/inotropes	138 (29.9)
Renal replacement therapy	51 (11.0)
Outcomes	
Decision to forgo life-sustaining therapies	164 (35.5)
ICU mortality	94 (20.3)
Length-of-stay in the ICU (days)	2 (1–5)
Hospital mortality	109 (23.6)
One-year mortality (available for 461 patients)	346 (73.0)

Continuous variables are expressed as median (interquartile range) and categorical variables as counts (percentages). ^a^ including urinary tract (n = 43, 9.3%), skin (n = 21, 4.5%), gynecologic (n = 17, 3.7%), breast (n = 15, 3.2%) and miscellaneous (n = 67, 14.5%). Abbreviations: ICU intensive care unit; SOFA Sequential Organ Failure Assessment.

**Table 2 cancers-14-05870-t002:** Functional and nutritional status.

Characteristics	n = 462 Patients
**At ICU admission**	
Functional status	
Performance status 3–4 (n = 381)	57 (15.0)
Nutritional status	
Body mass index (kg/m^2^) (n = 440)	23.2 (20.3–26.4)
Albumin level (g/L) (n = 392)	29 (24–34)
Prealbumin level (g/L) (n = 272)	0.14 (0.09–0.19)
Morphometric measurements (n = 290)	
Muscle area (cm^2^)	
Psoas area	12.9 (9.6–16.0)
Parietal area	101.4 (84.9–120.7)
Total muscle area	116.8 (95.5–136.1)
Skeletal muscle index (SMI) (cm^2^/m^2^)	
SMI psoas area	4.4 (3.8–5.9)
SMI parietal area	35.0 (30.0–41.2)
SMI total muscle area	39.3 (33.9–46.8)
SMI total muscle area < 55 cm^2^/m^2^ (180 male patients)	167 (92.8)
SMI total muscle area < 39 cm^2^/m^2^ (110 women patients)	74 (67.3)
**Six months before ICU admission**	
Functional status	
Performance status 3–4 (n = 275)	9 (3.3)
Nutritional status	
Body mass index (kg/m^2^) (n = 385)	24.4 (21.6–28.1)
Albumin level (g/L) (n = 376)	37 (33–41)
Prealbumin level (g/L) (n = 284)	0.20 (0.15–0.26)

Continuous variables are expressed as median (interquartile range) and categorical variables as counts (percentages). Numbers in brackets in the left column correspond to the number of patients with available data. Abbreviations: ICU intensive care unit; SMI skeletal muscle index.

**Table 3 cancers-14-05870-t003:** Factors associated with ICU mortality in the whole cohort and in the subgroup of patients undergoing invasive mechanical ventilation.

Characteristics	CSH [95% CI]	*p*
**Whole cohort** (**n = 462**)		
Surgery < 3 months	0.15 [0.04–0.58]	0.0006
Specific complications	2.09 [1.02–4.28]	0.04
SOFA score at admission	1.21 [1.11–1.31] ^a^	<0.001
Performance status at admission	1.74 [1.27–2.39] ^a^	<0.001
Albumin variation within the past 6 months		
Stable	Ref.	-
Decrease > 5%	-	-
Increase > 5%	0.38 [0.16–0.87]	0.02
**Invasive mechanical ventilation** (**n = 151**)		
Surgery < 3 months	0.16 [0.04–0.65]	0.01
Specific complications	2.38 [1.03–5.97]	0.04
SOFA score at admission	1.07 [1.01–1.15] ^a^	0.03
SMI psoas area	0.82 [0.67–0.98]	0.04

^a^ per point increase. Abbreviations: CSH cause specific hazard; SOFA Sequential Organ Failure Assessment; SMI skeletal muscle index.

**Table 4 cancers-14-05870-t004:** Determinants of one-year mortality in the 368 ICU survivors.

Characteristics	CSH [95% CI]	*p*
Type of cancer		
Non-lung cancer	Ref.	-
Lung cancer	1.84 [1.29–2.64]	<0.001
Status in progression	2.24 [1.57–3.18]	<0.001
Decision to forgo life-sustaining therapy	2.40 [1.65–3.50]	<0.001
Performance status at admission	1.34 [1.14–1.59] ^a^	<0.001
Weight variation within the past six months		
Stable	Ref	-
Decrease > 10%	1.33 [1.17–2.99]	0.01
Increase > 10%	0.88 [0.31–2.53]	0.82

^a^ per point increase. Abbreviations: CSH cause specific hazard.

## Data Availability

The datasets used and/or analyzed during the current study are available from the corresponding author on reasonable request.

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
