# Peer review of "Assessment of Functional and Nutritional Status and Skeletal Muscle Mass for the Prognosis of Critically Ill Solid Cancer Patients"

_cancers, 2022, doi:10.3390/cancers14235870_

Round 1
Reviewer 1 Report
[line# 134] Authors described that they used “performing stepwise backward and forward variable selections. The model included variables that reached a p value be-135 low 0.20 in univariate analysis.” Stepwise/backward/forward variable selections are three different methods used on variable selections. Did they provide the same result? Which method did you use? The variable selection method is not clear. You need to describe it clearly.
[line# 185] “Independent predictors of ICU mortality were the following (table 3):~” Did you present the multivariate Cox cause-specific model? The sentence needs to be more detail to lead clear understanding.
Author Response
[line# 134] Authors described that they used “performing stepwise backward and forward variable selections. The model included variables that reached a p value be-135 low 0.20 in univariate analysis.” Stepwise/backward/forward variable selections are three different methods used on variable selections. Did they provide the same result? Which method did you use? The variable selection method is not clear. You need to describe it clearly.
We apologize for the unclear sentence concerning the covariate selection method. We used a stepwise regression with bidirectional elimination approach which is essentially a forward selection procedure but with the possibility of deleting a selected variable at each stage, as in the backward elimination, when there are correlations between variables (Chatterjee, S.; Hadi, A.S. Regression Analysis by Example; Wiley series in probability and statistics; Fifth edition.; Wiley: Hoboken, New Jersey, 2012; ISBN 978-0-470-90584-5). Sentence was modified by "performing stepwise regression with bidirectional approach for the covariate selection procedure".
[line# 185] “Independent predictors of ICU mortality were the following (table 3):~” Did you present the multivariate Cox cause-specific model? The sentence needs to be more detail to lead clear understanding
Yes, we present the results of the multivariate Cox cause-specific model as suggested by the Cox cause-specific hazard reported in the Table 3. We modified the sentence for clarity.
Reviewer 2 Report
It is well conceivable that functional and nutritional status and sarcopenia can affect
the prognosis of critically ill solid cancer patients. However, I think that this manuscript has major flaws described as follows.
1. It seems that there are some misunderstandings about the basic concept of the relevant fields eg nutritional status, sarcopenia or cachexia.
Sarcopenia was first described as age related reduction of muscle mass however, it must be noted that recent definition of sarcopenia has been revised to "loss of muscle mass and muscle function" because evidence to data has suggested a stronger predictive association between muscle function than muscle mass and clinical outcomes.
It is well described in the textbooks and in many articles specialized in nutrition that serum albumin concentration can serves as a good indicator of prognosis but cannot be a marker for nutritional status.
2. It is difficult to find clinical relevance.
It is alluded in the introduction that the result of this study may help to decide whether patients should be admitted to the ICU or not. However, I do not think that it would do so because the most important factor in the decision making in this situation is the status of cancer itself. In this study, variety of patients with different cancer stages and prognosis are discussed altogether which make it difficult to interpret the results.
Author Response
It is well conceivable that functional and nutritional status and sarcopenia can affect the prognosis of critically ill solid cancer patients. However, I think that this manuscript has major flaws described as follows.
- It seems that there are some misunderstandings about the basic concept of the relevant fields eg nutritional status, sarcopenia or cachexia.
Sarcopenia was first described as age related reduction of muscle mass however, it must be noted that recent definition of sarcopenia has been revised to "loss of muscle mass and muscle function" because evidence to data has suggested a stronger predictive association between muscle function than muscle mass and clinical outcomes.
We apologize for this inaccurate definition. We modified the manuscript and added one reference to improve the clarity of those concepts.
It is well described in the textbooks and in many articles specialized in nutrition that serum albumin concentration can serves as a good indicator of prognosis but cannot be a marker for nutritional status.
We clarified this point adding a sentence in the discussion: “Moreover , serum albumin level is not an accurate and specific marker of nutritional status, since influenced by inflammation, intravascular fluid content, drugs and many conditions impacting its synthesis or loss”
- It is difficult to find clinical relevance.
It is alluded in the introduction that the result of this study may help to decide whether patients should be admitted to the ICU or not. However, I do not think that it would do so because the most important factor in the decision making in this situation is the status of cancer itself. In this study, variety of patients with different cancer stages and prognosis are discussed altogether which make it difficult to interpret the results.
We added the following sentences to the discussion section:
“The oncologic characteristics are known to have little impact on short-term survival though they account as important determinants of long-term outcomes [2,3]. The extent and reversibility of organ failures are hardly predictable before ICU admission since highly dependent on the definite diagnosis of the acute complication. Patients’ preferences regarding invasive management in the ICU are highly desirable but most often not expressed before, and unlikely to cover all situations encountered. Appraisal of functional and nutritional status appears prominent to the decision-making process.”
Reviewer 3 Report
The Authors evaluated the impact of functional and nutritional status and sarcopenia in adult patients with solid cancer requiring unplanned ICU admission. Performance status (PS), body weight and albumin level were collected at ICU admission and during the past six months. Sarcopenia was assessed at ICU admission by measurement of muscle areas normalized by height (SMI). 462 patients were analysed, mainly with gastro-intestinal (34.8%) and 35 lung (29.9%) neoplasms. 92.8% of men and 67.3% of women were deemed sarcopenic. In multivariate analysis, Performance Status at ICU admission and 6-month increase in albumin level were independent predictors of ICU mortality. In the subgroup of mechanically ventilated patients, psoas SMI was independently associated with ICU mortality. Among the 368 ICU-survivors, PS at ICU admission and 6-month weight loss were associated with one-year mortality.
I found the manuscript of interest. Methodology is appropriate. Results are well presented. Conclusions are supported by results. I do not have any suggestion to improve the manuscript
Author Response
The Authors evaluated the impact of functional and nutritional status and sarcopenia in adult patients with solid cancer requiring unplanned ICU admission. Performance status (PS), body weight and albumin level were collected at ICU admission and during the past six months. Sarcopenia was assessed at ICU admission by measurement of muscle areas normalized by height (SMI). 462 patients were analysed, mainly with gastro-intestinal (34.8%) and 35 lung (29.9%) neoplasms. 92.8% of men and 67.3% of women were deemed sarcopenic. In multivariate analysis, Performance Status at ICU admission and 6-month increase in albumin level were independent predictors of ICU mortality. In the subgroup of mechanically ventilated patients, psoas SMI was independently associated with ICU mortality. Among the 368 ICU-survivors, PS at ICU admission and 6-month weight loss were associated with one-year mortality.
I found the manuscript of interest. Methodology is appropriate. Results are well presented. Conclusions are supported by results. I do not have any suggestion to improve the manuscript.
We thank reviewer 3 for his/her encouraging comments.
Round 2
Reviewer 2 Report
The term “sarcopenia” is changed to “cachexia” in several places, yet sarcopenia is still used in the title and other places which is more confusing. In addition, it seems to me that authors do not fully understand the complex concept of cachexia. In this paper, you looked at the muscle area by CT to assess muscle mass. I would suggest that you simply use the term “reduced muscle mass” or “low muscle mass” instead of sarcopenia or cachexia.
Line 62
“nutritional status, which can be assessed by simple clinical and biological parameters. “
Is this true? What are you assuming as simple clinical and biological parameters? If you assume serum albumin levels as simple biological parameter, it should be reconsidered.
Line 63
“Thus cancer cachexia is defined by a body mass index (BMI) lower than 20 kg/m2.”
I don’t think that cancer cachexia is solely defined by a body mass index (BMI) lower than 20 kg/m2.
Line 110
You are not assessing sarcopenia by CT.
You evaluated the skeletal muscle mass.
Line 222
How does the result of this study specifically change decision making?
Line 235-239
“thereby reflecting impaired nutritional status but also likely related to chronic inflammation. Nevertheless, we found an association between 6-month increase in albumin level and ICU-survival, improvement in recent nutritional status could contribute to the prognosis”
This statement should be changed.
Roughly speaking, albumin indicates how sick the patient is. It may be especially true in cancer patients. Albumin can be used as a risk factor for low nutritional status. But cannot (should not) be used for assessing nutritional status or determining the effects of nutritional treatments.
Ref)
Franch-Arcas G: The meaning of hypoalbuminaemia in clinical practice. Clin Nutr 2001, 20:265-269.
Fuhrman MP: The albumin-nutrition connection: separating myth from fact. Nutrition 2002, 18:199-200.
Lee JL, Oh ES, Lee RW, et al: Serum Albumin and Prealbumin in Calorically Restricted, Nondiseased Individuals: A Systematic Review. Am J Med 2015, 128:1023 e1021-1022.
Line 249-257
“sarcopenia” and “cachexia” are mixed up.
Line 267-286
This part should be reconstructed.
Sarcopenia and muscle mass reduction is not the same.
I would like to emphasize that this paper did not assessed sarcopenia but evaluated the muscle mass.
Ref
Cruz-Jentoft AJ, Baeyens JP, Bauer JM, et al: Sarcopenia: European consensus on definition and diagnosis: Report of the European Working Group on Sarcopenia in Older People. Age Ageing 2010, 39:412-423.
Cruz-Jentoft AJ, Bahat G, Bauer J, et al: Sarcopenia: revised European consensus on definition and diagnosis. Age Ageing 2019, 48:16-31.
Barazzoni R, Jensen GL, Correia M, et al: Guidance for assessment of the muscle mass phenotypic criterion for the Global Leadership Initiative on Malnutrition (GLIM) diagnosis of malnutrition. Clin Nutr 2022, 41:1425-1433.
Author Response
Dear Reviewer,
We thank you for your careful reviews of our manuscript, for your constructive criticisms and suggestions to improve it.
We provide below a point-by-point reply, and we corrected the manuscript accordingly.
We hope that you will find the topic relevant for the readership of the Journal and that you will consider this revised version suitable for publication.
Sincerely yours,
Clara Vigneron and Frédéric Pène
The term “sarcopenia” is changed to “cachexia” in several places, yet sarcopenia is still used in the title and other places which is more confusing. In addition, it seems to me that authors do not fully understand the complex concept of cachexia. In this paper, you looked at the muscle area by CT to assess muscle mass. I would suggest that you simply use the term “reduced muscle mass” or “low muscle mass” instead of sarcopenia or cachexia.
We apologize for this confusion and we changed the term sarcopenia to “low or reduced muscle mass” throughout the manuscript,
For cachexia, we used the definitions proposed by Fearon et al (reference 20): Fearon, K.; Strasser, F.; Anker, S.D.; Bosaeus, I.; Bruera, E.; Fainsinger, R.L.; Jatoi, A.; Loprinzi, C.; MacDonald, N.; Mantovani, G.; et al. Definition and Classification of Cancer Cachexia: An International Consensus. Lancet Oncol 2011, 12, 489–495, doi:10.1016/S1470-2045(10)70218-7.
Line 62
“nutritional status, which can be assessed by simple clinical and biological parameters. “
Is this true? What are you assuming as simple clinical and biological parameters? If you assume serum albumin levels as simple biological parameter, it should be reconsidered.
We understand the reviewer’s concerns about the consideration of albumin as nutritional parameter. However, the level of serum albumin relies on nutritional status, but not only because of various alternative mechanisms resulting in low levels during acute and chronic inflammatory conditions. We discussed it in the discussion section, and remained very cautious in the interpretation of this variable. We removed from the introduction the sentence you mentioned.
Line 63
“Thus cancer cachexia is defined by a body mass index (BMI) lower than 20 kg/m2.”
I don’t think that cancer cachexia is solely defined by a body mass index (BMI) lower than 20 kg/m2.
We provided the recent definitions of cachexia as proposed by a panel of experts in the Material and Methods section (reference 20). This now reads as: Based on international expert consensus, lumbar SMI lower than 55 cm²/m² for men and 39 cm²/m² for women defined cachexia [20].
The extended definitions of cachexia were also mentioned in the discussion section: In 2011, a panel of experts used a Delphi process to propose different definitions of cancer cachexia, including BMI lower than 20 kg/m2, five-percent weight loss over the past 6 months, but also low lumbar skeletal muscle index (men <55 cm²/m²; women <39 cm²/m²).
Line 110
You are not assessing sarcopenia by CT. You evaluated the skeletal muscle mass
Thank you for this remark, we changed the term of sarcopenia to “low muscle mass”.
Line 222
How does the result of this study specifically change decision making?
The decision-making process of ICU admission and the extent of life-supporting therapies is complex and is based on multiple factors. Among them, the performance status is definitely a major clinical factor to be taken into account, but may result from several mechanisms. This exploratory study allows a better understanding of the determinants of the performance status in this particular population of cancer patients hospitalized in the ICU. We removed an awkward sentence in the introduction section and replaced it by the following (2nd paragraph): Functional status relies on disease stage but also on nutritional status and muscle preservation.
“thereby reflecting impaired nutritional status but also likely related to chronic inflammation. Nevertheless, we found an association between 6-month increase in albumin level and ICU-survival, improvement in recent nutritional status could contribute to the prognosis”
This statement should be changed.
Roughly speaking, albumin indicates how sick the patient is. It may be especially true in cancer patients. Albumin can be used as a risk factor for low nutritional status. But cannot (should not) be used for assessing nutritional status or determining the effects of nutritional treatments.
We understand your point and we improved the discussion section accordingly. We added one suggested reference.
“However, serum albumin level is neither an accurate nor a specific marker of nutritional status, since also influenced by inflammation, intravascular fluid content, drugs and many conditions impacting its synthesis or loss. Besides nutritional status, serum albumin level may thus rather account as a surrogate of general health condition.”.
Line 249-257
“sarcopenia” and “cachexia” are mixed up.
We clarified this section and we provided the definition of cachexia proposed by Fearon et al.
Line 267-286
This part should be reconstructed. Sarcopenia and muscle mass reduction is not the same.
I would like to emphasize that this paper did not assessed sarcopenia but evaluated the muscle mass.
We acknowledge that sarcopenia is not well-defined and we added the recent European consensus suggested by the reviewer.
We clarified this by suppressing the term of sarcopenia in the manuscript (except in one place to explain the concerns related to the unclear definition) and we generally used “low or reduced muscle mass” instead.
